# FEATURE PRIORITIZATION AND REGULARIZATION IMPROVE STANDARD ACCURACY AND ADVERSARIAL ROBUSTNESS

## ABSTRACT

Adversarial training has been successfully applied to build robust models at a certain cost. While the robustness of a model increases, the standard classification accuracy declines. This phenomenon is suggested to be an inherent trade-off. We propose a model that employs feature prioritization by a nonlinear attention module and $L_2$ feature regularization to improve the adversarial robustness and the standard accuracy relative to adversarial training. The attention module encourages the model to rely heavily on robust features by assigning larger weights to them while suppressing non-robust features. The regularizer encourages the model to extracts similar features for the natural and adversarial images, effectively ignoring the added perturbation. In addition to evaluating the robustness of our model, we provide justification for the attention module and propose a novel experimental strategy that quantitatively demonstrates that our model is almost ideally aligned with salient data characteristics. Additional experimental results illustrate the power of our model relative to the state of the art methods.

## 1 INTRODUCTION

Deep learning models have demonstrated impressive performance in a wide variety of applications (Goodfellow et al., 2016; Krizhevsky et al., 2012; Simonyan & Zisserman, 2014; Mnih et al., 2015). However, recent works have shown that these models are susceptible to adversarial attacks: imperceptible but carefully chosen perturbation added to the input can cause the model to make highly confident but incorrect predictions (Szegedy et al., 2013; Goodfellow et al., 2015; Kurakin et al., 2016).

Exploring the adversarial robustness of neural networks has recently gained significant attention and there is a rapidly growing body of work related to this topic (Kurakin et al., 2016; Tramèr et al., 2017; Fawzi et al., 2018; Athalye & Sutskever, 2017; Carlini & Wagner, 2017; Kolter & Wong, 2017; Wong & Kolter, 2018; Madry et al., 2017). A wide variety of methods are proposed to defend a model against adversarial attacks (Prakash et al., 2018; Liao et al., 2018; Song et al., 2017; Samangouei et al., 2018). Despite these advances, many techniques are subsequently shown to be ineffective (Athalye et al., 2018; Athalye & Carlini, 2018), and adversarial training which uses adversarial samples in addition to clean images during the training process has been shown to be able to build relatively robust neural networks (Madry et al., 2017; Athalye et al., 2018; Dvijotham et al., 2018). With strong adversaries such as the Projected Gradient Descent (PGD) (Madry et al., 2017) or the Iterative Fast Gradient Sign Method (I-FGSM) (Kurakin et al., 2016) adversarially trained models are able to achieve state-of-the-art performance against a wide range of attacks.

Recent advances in the understanding of adversarial training provide insights of its effectiveness. It is shown that standard and robust models depend on very different sets of features (Tsipras et al., 2018; Tanay et al., 2018). While standard models utilize features including non-robust ones that are weakly correlated with class labels and easily manipulated by small input perturbations, robust models only use robust features that are highly correlated with class labels and invariant to those perturbations. Although adversarial training learns robust features, there are no explicit design components to encourage a model to depend solely on robust features. Therefore, to further improve the robustness of a model, we propose feature regularization and prioritization schemes.

We first propose to use attention mechanism to bias a model towards robust features that are highly correlated with the class labels. We call the learned features at the final layer of a network the global features, and the ones at lower level layers the local features. In our attention module, we use the global features as a way to assign weights to the local features by a non-linear compatibility function. Since global features are directly used to produce class label prediction, we are effectively assigning weights to local features depending on their correlation to the label. Robust features have higher correlation and therefore will be assigned larger weights which in turn contribute to the model's robustness.

Next, we propose to use feature regularization to learn robust features that are invariant to input perturbations. We add a $L_2$ regularization term that penalizes the distance between the learned features of a clean sample $x$ and that of its perturbed adversarial counterpart $x'$ to the training objective. By optimizing this regularizer, we are pushing the model to extract very similar features from the original image and the adversarial image, thus only features that are invariant to the perturbations are learned and we effectively ignore the added noise. From another point of view, a model with small $L_2$ feature distance maps the two nearby points in the image space to nearby points in the learned high dimensional manifold which is a desirable behavior.

In this paper, we propose an approach that enhances adversarial training with feature prioritization and regularization to improve the robustness of a model. We use extensive experiments to demonstrate that the attention module focuses on the area of an image which contains the actual object and helps the classifier to only rely on features extracted from those areas. The background clutter and irrelevant features which could be misleading are suppressed. The feature regularization further encourages the model to extract robust features that are not manipulated by the adversarial perturbations. The resulting model has a highly interpretable gradient map that aligns perfectly with salient data characteristics.

The main contributions of this paper include:

- A method based on feature prioritization and regularization, which significantly outperforms adversarial training. Our model is evaluated on the MNIST, CIFAR-10, and CIFAR-100 datasets, and demonstrates superior performance relative to both standard classification accuracy and adversarial robustness.

- We provide justification to show that the attention module helps the model to rely on robust features by assigning larger weights to them. Through qualitative inspection, we show that the attention maps generated by our non-linear attention estimator focus sharply on the regions of interest while suppressing irrelevant background clutter.

- In addition to qualitative evaluation of the gradient maps, we propose a novel experimental strategy that quantitatively demonstrates the better alignment of the gradient maps generated by our model with salient data characteristics.

## 2 RELATED WORK

Due to the extensive amount of literature in this area and the limited length of this paper, we only review some of the most related works in this section. For a more comprehensive survey please refer to Akhtar & Mian (2018).

*Adversarial training.* Kurakin et al. (2016) use adversarial training as a form of data augmentation where it injects adversarial examples during training. In every training mini batch, a mixture of clean images and adversarial images generated by one step Fast Gradient Sign Method (FGSM) are used to update the network's parameters. It is then improved by Na et al. (2017) by adding adversarial examples generated by iterative methods. In Madry et al. (2017), it is proposed to replace all clean images with adversarial images which is a direct result of optimizing a saddle point (min-max) formulation. By studying the loss landscape of the problem, they suggest that PGD is a universal first-order adversary which is then used in their adversary generating process.

*$L_2$ regularization.* A similar idea with feature regularization is proposed in Kannan et al. (2018) which they call logit pairing, to prevent a model from being over-confident when making predictions. Compared with logit pairing, feature regularization is more intuitive as it motivates a model to learn very robust features that are invariant to input perturbations, which leads to a robust model. In

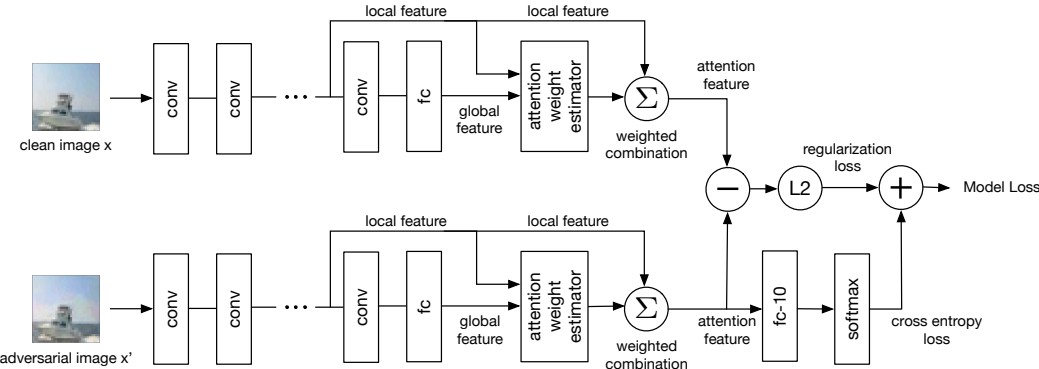

Figure 1: Overview of the proposed model. The top and bottom networks are the same copy that share all network parameters. Both the clean and adversarial images are forwarded through the network to produce the corresponding attention features. The $L_2$ regularization loss is defined as the Euclidean distance between the two sets of attention features. The final model loss is a combination of the $L_2$ regularization loss and the cross-entropy loss for only the adversarial input.

addition, we propose to also use attention module to further encourage the model to favor robust features which will improve the robustness.

*Attention Models.* Attention in CNN is most commonly deployed for query-based tasks (Seo et al., 2016; Xu et al., 2015; Jetley et al., 2018). In Jetley et al. (2018) a method is presented to use a learned representation of the global image as a query to leverage multiple attention maps at different scales, which allows the expression of a complementary focus on different parts of the image. However, the application of attention to the adversarial robustness aspects has not been seriously explored. To the best of our knowledge, we are the first to employ an attention mechanism in training a robust deep neural network. In our application, we use a ReLU activated neural network instead of the linear-based method as the attention estimator. It allows highly non-linear compatibility between the learned global features and the lower-level local features.

## 3    APPROACH

We now present our model that combines the attention module and $L_2$ feature regularization, and show how it can be applied to enhance the adversarial training to improve the adversarial robustness of a model and its accuracy. Figure 1 provides an overview of our method. We adopt the terminology in Jetley et al. (2018), in particular that local features and global features refer to features extracted at certain layers of a network whose effective receptive fields are subsets of the image and the entire image. We start by forwarding each of the clean and adversarial images and computing the attention weights by a non-linear estimator. Then the individual attention feature is defined to be the weighted combination of the corresponding local features. Next, we define an $L_2$ regularization loss to be the Euclidean distance between the two sets of learned attention features. The attention features of the adversarial image are then used to produce the logits, which is followed by softmax layer to produce the cross-entropy loss. The final loss function of our model is a combination of cross-entropy loss and the regularization loss.

### 3.1    ADVERSARIAL TRAINING

We adopt the adversarial training described in Madry et al. (2017) as the basic training approach. It replaces natural training examples by PGD examples, which is suggested to represent a universal first-order adversary. So far PGD has been shown to represent the strongest attack method (Athalye et al., 2018; Athalye & Carlini, 2018). A model that is trained with PGD adversaries is also robust against a wide range of other attacks and not yet outperformed by any other approach. The adversarial training has a saddle point formulation:

$$\min_{\theta} \mathbb{E}_{(\boldsymbol{x},y)\sim\mathcal{D}}[\max_{\boldsymbol{\delta}\in S} L(\theta, \boldsymbol{x} + \boldsymbol{\delta}, y)] \tag{1}$$

where $\mathcal{D}$ is the distribution of data $\boldsymbol{x}$ and class labels $y$, $L$ is the cross-entropy loss function for a model with parameters $\theta$, $\boldsymbol{\delta}$ is the additive adversarial perturbation with bound $S$. In this paper we consider $l_\infty$ bound as in Madry et al. (2017). Our adversarial samples $\boldsymbol{x}' = \boldsymbol{x} + \boldsymbol{\delta}$ are created by PGD:

$$\boldsymbol{x}^{t+1} = \Pi_{\boldsymbol{x}+S}\left(\boldsymbol{x}^t + \alpha\,\text{sgn}(\nabla_{\boldsymbol{x}}L(\theta, \boldsymbol{x}, y))\right) \tag{2}$$

PGD adversaries are computed at each iteration as an approximated optimum of the inner maximization in equation (1) and an update of the parameters $\theta$ is made according to the outer minimization formulation.

## 3.2 ATTENTION MODEL

As we discussed in Section 1, our goal of attention model is to favor robust features in making predictions. We propose a non-linear attention model that acts as a feature prioritizing scheme, which is able to put more weight on robust features and less weight on non-robust features to increase the robustness of a classifier.

Let $\boldsymbol{l}_n^i$ be the learned feature vector at layer $i \in \{1, 2, ..., I\}$ of a neural network at spatial location $n \in \{1, 2, ..., N\}$, and let $\boldsymbol{g}$ be the feature vector of the layer just before the final fully connected layer which produces the class label prediction scores (logits). We use a small ReLU activated neural network to generate compatibility scores between the global feature $\boldsymbol{g}$ and local features $\boldsymbol{l}_n^i$:

$$c_n^i = f(\boldsymbol{l}_n^i, \boldsymbol{g}) \tag{3}$$

where $f$ is the neural network and the concatenation of $\boldsymbol{g}$ and $\boldsymbol{l}_n^i$ is fed to the network to produce the compatibility scores $c_n^i$. We then normalize the scores with a softmax operation to get the attention weights:

$$w_n^i = \frac{\exp c_n^i}{\sum_m \exp c_m^i} \tag{4}$$

Next, we compute the weighted sum of local feature vectors which is the attention feature vector at layer $i$:

$$\boldsymbol{h}^i = \sum_n w_n^i \boldsymbol{l}_n^i \tag{5}$$

We use the outputs of the last residual block as the local feature for computing attention, and replace the global feature $\boldsymbol{g}$ with the corresponding attention descriptor $\boldsymbol{h}^i$ for final classification.

By using a small neural network instead of the linear alignment models as in Jetley et al. (2018), we are able to capture non-linear compatibility between the local and global features when producing the attention weights, which is beneficial considering the multiple non-linear function activated layers between the local and global features.

## 3.3 FEATURE REGULARIZATION

In addition to the attention mechanism, we also propose to use an $L_2$ regularization term to encourage the model to extract similar features for the clean image and the corresponding adversarial image. Denote by $\mathcal{G}_\theta$ the deep neural network, $\boldsymbol{x}$ and $\boldsymbol{x}'$ the natural image and adversarial image. Denote by $\mathcal{G}_\theta(\boldsymbol{x})$, $\mathcal{G}_\theta(\boldsymbol{x}')$ the learned features of the layer just before the final fully connected layer (in our case this is the attention weighted global descriptor) which produces the class label prediction scores. The $L_2$ regularizer has the following form:

$$L_r(\boldsymbol{x}, \boldsymbol{x}') = \|\mathcal{G}_\theta(\boldsymbol{x}) - \mathcal{G}_\theta(\boldsymbol{x}')\|_2 \tag{6}$$

By minimizing the regularization function, the model effectively learns very similar features for the clean sample and the adversarial sample, which are robust features since they are invariant to the adversarial perturbation. From another perspective, the learned features of a neural network lie on a high dimensional manifold that is linearly separable for different classes because the classification layer is a linear classifier followed by a softmax function. With adversarial training alone, a model

only tries to map $\boldsymbol{x}$ and $\boldsymbol{x}'$ to the same side of the decision boundary, while with the additional regularization, they are not only on the same side but also mapped to nearby points in the space. This mapping is a desired behavior considering that, in the original image space, they are very close points representing essentially the same image.

### 3.4 MODEL LOSS

Equipped with the presented methods, the total loss of our model is:

$$\text{Loss} = \mathbb{E}_{(\boldsymbol{x},y)\sim\mathcal{D}}[L(\theta, \boldsymbol{x}', y) + \lambda\|\mathcal{G}_\theta(\boldsymbol{x}) - \mathcal{G}_\theta(\boldsymbol{x}')\|_2] \tag{7}$$

where $\lambda$ is a hyperparameter that controls the relative weight of the $L_2$ regularization loss.

## 4 EXPERIMENTS AND RESULTS

In this section, we evaluate our model on the MNIST, CIFAR-10 and CIFAR-100 datasets, and present justification to attention module and some quantitative and qualitative results.

### 4.1 ROBUSTNESS ON MNIST

First we present the results on the MNIST dataset. We use a CNN with two convolutional layers with 32 and 64 filters respectively, each followed by $2\times2$ max-pooling, and a fully connected layer of size 1024. For the PGD adversary, we run 40 iterations with a step size of 0.01 and $l_\infty$ bound of $\epsilon = 0.3$. The settings are the same as in Madry et al. (2017). Since MNIST is a very small scale dataset and the model is very robust with just adversarial training, we do not employ the attention mechanism on MNIST. We only study the effectiveness of the proposed feature regularization method.

Table 1: Performance comparison of the adversarial training method (Madry et al., 2017) and our proposed adversarial training with feature regularization (AT-reg) method on MNIST. Black box accuracies are evaluated against adversaries generated from an independently trained copy of the same method with identical configurations.

| Method | Natural | White box | Black box |
|---|---|---|---|
| Madry et al. (2017) | **98.72%** | 92.86% | 95.97% |
| AT-reg | 98.66% | **95.69%** | **96.90%** |

The evaluation results on MNIST are presented in Table 1. Regarding the value of weight $\lambda$ of the $L_2$ regularizer, we find that roughly any $\lambda \in [0.001, 0.05]$ works well. The reported results are obtained with $\lambda = 0.01$. From Table 1, we can see that a model trained with the proposed feature regularization method is significantly more robust against PGD adversary than the baseline model that uses adversarial training alone. The improvement is nearly 3% for white box attack and 1% for black box attack.

### 4.2 ROBUSTNESS ON CIFAR-10

Here we present our results on the CIFAR-10 dataset. For the base network, we use the original ResNet model that has four residual blocks with [16, 16, 32, 64] filters and its 3-times wider variant with [16, 48, 96, 192] filters respectively. For our attention model, we modify the ResNet by replacing the spatial global average pooling layer after the residual block 4 with a convolutional layer sandwiched between two max-pooling layers to obtain the global feature $\boldsymbol{g}$. We use a one-hidden-layer neural network with 64 hidden units and ReLU activation function as the non-linear attention weight estimator. For the PGD adversary, we run 5 iterations with a step size of 2 and $l_\infty$ bound of $\epsilon = 8/255$. In order to isolate and analyze the effectiveness of attention module and implicit denoising independently, we train three models with the following configurations: AT-reg is an adversarial trained model with feature regularization, AT-att is an adversarial trained model with attention module, and AT-att-reg is the model with both attention and feature regularization.

Table 2: Performance comparison of the adversarial training (Madry et al., 2017), adversarial training with feature regularization (AT-reg), adversarial training with attention model (AT-att), and adversarial training with both (AT-att-reg) on CIFAR-10. Black box accuracies are evaluated against adversaries generated from an independently trained copy of the same method with identical configurations.

| Method | Madry et al. (2017) | AT-reg | AT-att | AT-att-reg |
|---|---|---|---|---|
| Natural | 80.79% | 79.52% | **82.75%** | 81.21% |
| White box, PGD 5 steps | 49.89% | 52.35% | 51.20% | **52.81%** |
| White box, PGD 20 steps | 39.72% | 44.25% | 41.39% | **44.58%** |
| White box, PGD 100 steps | 38.76% | 43.73% | 40.47% | **44.04%** |
| White box, PGD 200 steps | 38.64% | 43.70% | 40.36% | **44.02%** |
| White box, CW 30 steps | 40.27% | 42.96% | 40.60% | **43.09%** |
| White box, CW 100 steps | 39.98% | **42.87%** | 40.31% | 42.75% |
| Black box, PGD 5 steps | 60.13% | 61.82% | **63.26%** | 62.32% |
| Black box, PGD 20 steps | 56.60% | 56.40% | **58.26%** | 57.52% |
| Black box, PGD 100 steps | 56.44% | 56.28% | **58.08%** | 57.53% |
| Black box, PGD 200 steps | 56.49% | 56.25% | **58.04%** | 57.53% |
| Black box, CW 30 steps | 57.11% | 56.86% | **58.07%** | 57.57% |
| Black box, CW 100 steps | 57.10% | 56.79% | **58.13%** | 57.41% |

Table 3: Performance comparison of the adversarial training (Madry et al., 2017), adversarial training with feature regularization (AT-reg), adversarial training with attention model (AT-att), and adversarial training with both (AT-att-reg) on CIFAR-10 using the 3-times wide ResNet network. Black box accuracies are evaluated against adversaries generated from an independently trained copy of the same method with identical configurations.

| Method | Madry et al. (2017) | AT-reg | AT-att | AT-att-reg |
|---|---|---|---|---|
| Natural | 85.41% | 84.65% | **86.48%** | 85.98% |
| White box, PGD 5 steps | 49.15% | 52.21% | 50.91% | **53.23%** |
| White box, PGD 20 steps | 38.19% | 41.00% | 39.52% | **41.55%** |
| White box, PGD 100 steps | 37.39% | 40.28% | 38.98% | **40.78%** |
| White box, PGD 200 steps | 37.20% | 40.24% | 38.89% | **40.67%** |
| White box, CW 30 steps | 38.92% | **42.20%** | 40.75% | 42.12% |
| White box, CW 100 steps | 38.71% | 41.88% | 40.32% | **42.06%** |
| Black box, PGD 5 steps | 68.03% | 68.00% | **69.78%** | 69.00% |
| Black box, PGD 20 steps | 62.77% | 63.14% | **64.70%** | 64.01% |
| Black box, PGD 100 steps | 62.65% | 63.12% | **64.78%** | 64.15% |
| Black box, PGD 200 steps | 62.73% | 63.12% | **64.69%** | 64.11% |
| Black box, CW 30 steps | 63.89% | 64.23% | **65.65%** | 64.50% |
| Black box, CW 100 steps | 63.77% | 64.04% | **65.43%** | 64.35% |

The evaluation results of aforementioned four models on CIFAR10 are presented in Table 2. Similar with MNIST, we find that roughly any $\lambda \in [0.01, 0.1]$ works well for feature regularization. The reported results are obtained with $\lambda = 0.1$ for AT-reg and $\lambda = 0.01$ for AT-att-reg. From the table, we see that all of the three proposed models have better adversarial robustness over the baseline model that only uses adversarial training, and both models with attention show improvement on the classification accuracy on natural examples as well.

By comparing AT-reg with adversarial training (Madry et al., 2017), we note that the feature regularization method provides a significant improvement on white box robustness over the baseline method, with a cost of standard accuracy and a slight decline of black box accuracy. Same trade-off appears in the comparison of AT-att-reg and AT-att.

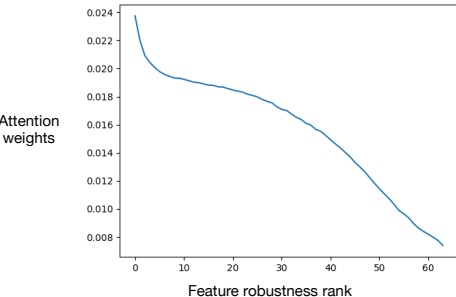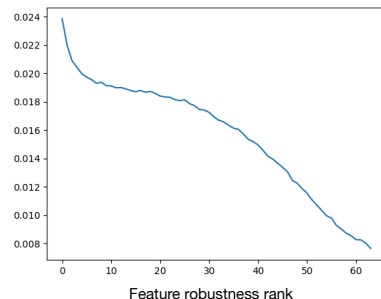

Figure 2: The relationship between attention weights and feature robustness. The horizontal axis is the robustness rank, with 0 being the most robust and 63 the least robust, and the vertical axis is the corresponding attention weights. Left plot shows the results for training set and right plot is for test set of CIFAR-10.

Next, by comparing the results of models with and without the attention module, we can see that attention contributes to both standard and adversarial accuracy. The attention structure not only favors robust features, it also relies heavily on features extracted from the spatial area that contains the actual object of concern. By suppressing the features extracted from the background clutter and misleading perturbations in irrelevant areas, the model with attention module more precisely learns the underlying distribution of the data therefore the better accuracy. In the next section, we will demonstrate that the attention module assigns larger weights to more robust features, and attention maps sharply focus on the object in the image and ignore the irrelevant background clutter.

### 4.3 JUSTIFICATION FOR ATTENTION

Now we investigate the attention module and demonstrate how it actually helps to train a more robust model. As we discussed in Section 1, the intuition for using attention is to assign larger weights on more robust features and smaller weights on less robust features. We now show that this is in fact true by examining the attention weights relative to the feature robustness.

Figure 2 shows the relationship between the robustness of a feature and the magnitude of its assigned attention weight. The robustness measure we use is the $L_2$ distance between the learned features of a clean and an adversarial image, i.e. the smaller the distance between the features, the more invariant the feature is against input perturbations, therefore the feature is more robust. For each plot in Figure 2, the horizontal axis is the rank of feature robustness, with 0 being the most robust and 63 the least robust, and the vertical axis is the corresponding attention weights. The plot on the left shows the average attention weights for all training images and the plot on the right is for all test images in CIFAR-10. We can see that our proposed attention mechanism indeed assigns larger weights to more robust features and less weights to non-robust features, so the model is more invariant to adversarial perturbation.

Next we show the attention maps of our model in Figure 3 to visualize the attention weights. The attention maps focus sharply on the objects in the images, and the most relevant features like the head and legs of an animal and the wings of an airplane contribute more to the model's prediction.

### 4.4 GRADIENT MAP

In this section we study the gradient maps, which are the gradients of the cross-entropy loss with respect to the input image pixels, for both the baseline and our model on CIFAR-10. The gradient maps are a direct indicator on how the input features are utilized by a model to produce a final prediction. A large magnitude of the gradient on an input feature signifies a heavy dependence the model has. Human vision is robust against small input perturbations and the perception of an image reflects which input features contribute to the human vision robustness. At the same time, the gradient maps of a robust model also highlight the input features which affect the loss most strongly, therefore more robust models depend on robust features and will be better aligned with human

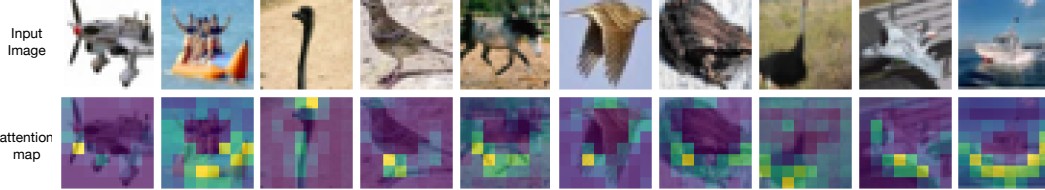

Figure 3: The learned attention maps of our model. The first row are the input images and the second are the attention maps learned at residual block 4.

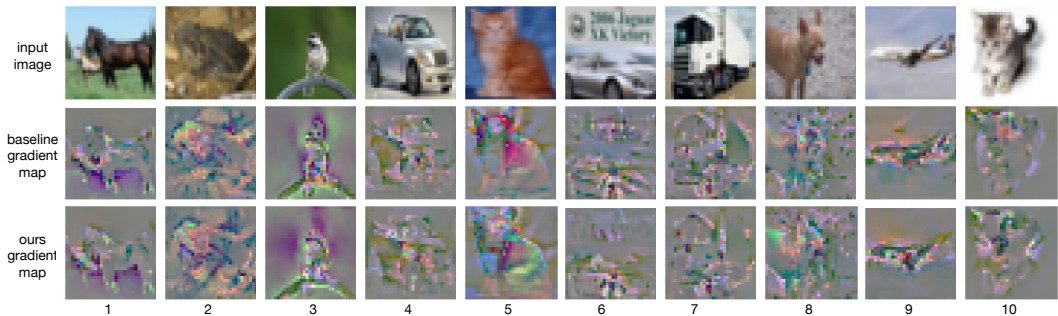

Figure 4: Gradient maps generated by compute gradient of model's cross-entropy loss with respect to the input images. Top row are the original CIFAR-10 images, midrow are the gradient maps of Madry et al. (2017), and bottom row are the gradient maps of our model. The raw gradients are clipped to within $\pm 3$ standard deviation and rescaled to lie in the [0, 1] range for visualization. No other preprocessing is applied.

vision. So the alignment of gradient maps with the image can be used to evaluate the robustness a model. Next, we show that the gradient maps generated from our model align better with the salient data characteristics by evaluating the gradient maps both qualitatively and quantitatively.

First we present the qualitative result. Figure 4 contains the gradient maps from Madry et al. (2017) and our model. These are raw gradients with only being clipped and rescaled for visualization. Overall, we note that both models generate highly interpretable gradient maps that align very well with the image features. However, upon carefully inspecting the images, it is evident that the gradient maps generated from our model are better than the baseline model. To point out a few: in columns 2, 3, 6 and 9, our gradient maps have cleaner backgrounds and the gradients only focus on the objects; especially in column 6, the baseline model has large gradients on the text field in the background which is irrelevant to the class label (automobile), while in our model gradients in that area are much more suppressed. In columns 1, 5, and 10 the edges of the faces and heads of the animals are depicted clearer in our model.

Next, we introduce a quantitative evaluation method for gradient maps. The problem we consider here is to decide how well the gradient maps align with the original images. The better they align, the more recognizable the gradient images are. Therefore, in addition to human inspection which could be very subjective, we propose to use a pretrained neural network to classify the gradient maps for all images in the dataset, both the training set and the test set. A standard neural network extracts relevant features from the inputs and make predictions based on the features. When a gradient map is highly aligned with the original image, the neural net is able to identify more relevant features and thus the classification accuracy will be higher. Therefore, with the classification accuracy of gradient maps from both models of all images we are able to quantitatively compare the alignment.

The pretrained model we use is the same ResNet model as in Section 4.2 trained with only natural training data of CIFAR-10. It achieves an accuracy of 88.79% on the test set. The classification results are presented in Table 4. To avoid the possible influence of gradient clipping we also run the

evaluation on raw gradients. As demonstrated by the classification accuracy, the gradient maps from our model express significantly better alignment with the original images.

Table 4: Classification accuracy on the gradient maps from baseline and our methods on both the training dataset and test dataset of CIFAR-10. We run the experiment on gradient maps both with and without clipping to avoid the influence of gradient clipping.

| | With clipping | | Without clipping | |
|---|---|---|---|---|
| **Method** | **Train data** | **Test data** | **Train data** | **Test data** |
| Madry et al. (2017) | 27.10% | 26.78% | 28.60% | 28.72% |
| Ours | **30.11%** | **30.32%** | **31.46%** | **31.59%** |

To summarize, both the qualitative and quantitative results show that the gradient maps from our model have better interpretability and alignment with the original images. It suggests that our model depends on features that are very closely correlated with the robust features of the input images which explains the improved performance on both standard accuracy and adversarial robustness.

## 4.5 RESULTS ON CIFAR-100

Here we present our results on the CIFAR-100 dataset. The experiment setup is the same as CIFAR-10 in Section 4.2.

Table 5: Performance comparison of the adversarial training (Madry et al., 2017), adversarial training with feature regularization (AT-reg), adversarial training with attention model (AT-att), and adversarial training with both (AT-att-reg) on CIFAR-100. Black box accuracies are evaluated against adversaries generated from an independently trained copy of the same method with identical configurations.

| Method | Madry et al. (2017) | AT-reg | AT-att | AT-att-reg |
|---|---|---|---|---|
| Natural | 52.70% | 49.53% | **53.67%** | 50.66% |
| White box, PGD 5 steps | 25.14% | 26.99% | 26.33% | **27.76%** |
| White box, PGD 20 steps | 19.65% | 23.16% | 20.82% | **23.80%** |
| White box, PGD 100 steps | 19.47% | 23.07% | 20.59% | **23.62%** |
| White box, PGD 200 steps | 19.41% | 22.96% | 20.53% | **23.62%** |
| White box, CW 30 steps | 18.64% | 20.78% | 19.39% | **20.88%** |
| White box, CW 100 steps | 18.61% | 20.63% | 19.26% | **20.76%** |
| Black box, PGD 5 steps | 35.37% | 35.10% | **35.95%** | 35.17% |
| Black box, PGD 20 steps | 31.99% | 31.88% | **32.48%** | 32.04% |
| Black box, PGD 100 steps | 32.03% | 31.84% | **32.38%** | 32.00% |
| Black box, PGD 200 steps | 32.00% | 31.80% | **32.37%** | 32.06% |
| Black box, CW 30 steps | 32.50% | 31.96% | **32.75%** | 29.81% |
| Black box, CW 100 steps | 32.46% | 31.86% | **32.74%** | 29.71% |

## 5 CONCLUSION

In this paper we propose feature prioritization and regularization to enhance the adversarial robustness. With the non-linear attention module and $L_2$ feature regularization, a model is improved on both the standard classification accuracy and the adversarial robustness over the baseline adversarial training approach. We provide additional justifications for the attention module to show that it effectively favors robust features, and study the attention maps to demonstrate that the attention maps focus sharply on the region of interest. We then conduct quantitative and qualitative evaluation on gradient maps and show that they align perfectly with salient data characteristics, and therefore our model heavily relies on the robust features.

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
