# OpenReview forum: "Feature prioritization and regularization improve standard accuracy and adversarial robustness"
_ICLR.cc/2019/Conference_

### Official Review · AnonReviewer3 · 2018-11-01
**Interesting paper**

**Rating:** 5
**Confidence:** 2

**Review:**

This paper proposes a new architecture for adversarial training that is able to improve both accuracy and robustness performances using an attention-based model for feature prioritization and L2 regularization as implicit denoising. The paper is very clear and well written and the contribution is relevant to ICLR.

Pros:

- The background, model and experiments are clearly explained. The paper provides fair comparisons with a strong baseline on standard datasets.
- Using attention mechanisms to improve the model robustness in an adversarial training setting is a strong and novel contribution
- Both quantitative and qualitative results are interesting.

---

> ### Author Response · Authors · 2018-11-27
> **Response to Reviewer 3**
>
> We thank the reviewer for the kind comments.

---

### Official Review · AnonReviewer1 · 2018-11-01
**FEATURE PRIORITIZATION AND REGULARIZATION IMPROVE STANDARD ACCURACY AND ADVERSARIAL ROBUSTNESS**

**Rating:** 4
**Confidence:** 3

**Review:**

Summary: This paper argues that improved resistance to adversarial
attacks can be achieved by an implicit denoising method in which model
weights learned during adversarial training are encouraged to stay
close to a set of reference weights using the ell_2
penalty. Additionally, the authors claim that by introducing an
attention model which focuses the model training on more robust
features they can further improve performance. Some experiments are
provided.

Feedback: My main concerns with the paper are:

* The experimental section is fairly thin. There are at this point a
  large number of defense methods, of which Madry et al. is only one. In
  light of these, the experimental section should be expanded. The
  results should ideally be reported with error bars, which would help
  in gauging significance of the results.

* The differential impact of the two contributions is not entirely
  clear. The results in Table 1 suggest that implicit denoising can
  help, yet at the same time, Table 2 suggests that Black-box
  performance is better if we just use the attention model. Overall,
  this conflates the contributions unnecessarily and makes it hard to
  distingish their individual impact.

* The section on gradient maps is not clear. The authors argue that if
  the gradient map aligns with the image the model depends solely on
  the robust features. While this may be (somewhat more) intuitive in
  the context of simple GLMs, it's not clear why it should carry over
  to DNNs. I think it would help to make these intuitions much more
  precise. Secondly, even if this were the case, the methodology of
  using a neural net to classify gradient maps and from this derive a
  robustness metric raises precisely the kinds of robustness questions
  that the paper tries to answer. I.e.: how robust is the neural net
  classifying the gradient images, and how meaningful are it's
  predictions when gradient maps deviate from "clean" images.

Overall, I feel this paper has some potentially interesting ideas, but
needs additional work before it is ready for publication.

---

> ### Author Response · Authors · 2018-11-27
> **Response to Reviewer 1**
>
> We address the comments of this reviewer as follows.
>
> 1. We have run a significant amount of additional experiments and our proposed method demonstrates a consistent improvement over the baseline method. We agree that comparing with other defense methods would improve the experiment section, but we think that the performance comparison with Madry et al. (2017) is the most important for this work. Our feature prioritization and regularization techniques are used as an improvement over the baseline adversarial training approach. Adversarial training with PGD adversary is the state of art method which is validated in various papers and shows superior performance over other defense methods. Since we show that our methods work better than Madry et al. (2017) in various settings, the advantages transfer to other defense models.
>
> 2. With the results from extended experiments, we summarize the two contributions as follows: feature regularization significantly improves the white box robustness at the cost of a decline in standard accuracy, and a slight decline in black box accuracy; Attention improves both the white box and black box robustness, and at the same time also increases standard accuracy of a model.
>
> 3. The gradient maps are direct indicators on how the input features are utilized by a model to produce a final prediction. A large magnitude of the gradient on an input feature signifies a heavy dependence the model has. Human vision is robust against small input perturbations and the perception of an image reflects which input features contribute to the vision robustness. At the same time, the gradient maps of a model also highlight the input features which affect the loss most strongly, therefore more robust models depend on robust features and will be better aligned with human vision. So the alignment of gradient maps with the image can be used to evaluate the robustness a model. Next, the purpose of classifying gradient maps is to provide a comparable quantitative measure of the relative alignment between the two sets of gradient maps and the original images. A standard neural net extracts relevant features from the inputs and makes predictions based on the features. When a gradient map is highly aligned with the original image, the neural net is able to identify more relevant features and thus the classification accuracy will be higher. We agree that the robustness questions arise regarding the meaningfulness and interpretability of the accuracies, but we think this method works for comparison purposes.

---

### Official Review · AnonReviewer2 · 2018-11-02
**Lack of valid explanation, and insufficient experiment**

**Rating:** 5
**Confidence:** 5

**Review:**

This paper studies adversarial training of robust classification models. It is based on PGD training in [madry17]. It proposes two points: 1) add attention schemes, 2) add a feature regularization loss. The results on MNIST and CIFAR10 demonstrate the effectiveness. At last, it did some diagnostic study and visualization on the attention maps and gradient maps.

1. Can you provide detailed explanations/intuitions why attention will help train a more robust models?

2. Two related adversarial training papers are missing "Ensemble Adversarial Training" (ICLR2018) and "Adversarial Logit Pairing" (ICML2018). Also, feature (logit) regularization has been studied in ALP paper on ImageNet.

3. For Table 2 on CIFAR10, I would like to see PGD20 (iterations) + 2 (step size in pixels), PGD100 + 2 and PGD200 + 2. Also, I am interested in seeing CW loss which is based on logit margin.

4. I would like to see results using the "wide" model in [madry17] paper for ALP and LRM. I think results from large-capacity models are more convincing.

5. I would like to see results on CIFAR100, which is a harder dataset, 100 classes and 500 images per class. I think CIFAR10 alone is not sufficient for justification nowadays (maybe enough one year ago). Since ImageNet is,  to some extent, computationally impossible for schools, I want to see the justification results on CIFAR100.

##### Post-rebuttal

I appreciate the additional results in the rebuttal. I raise the score but it is still slightly below the acceptance. The reasons are 1) incremental novelty; 2) insufficient experiments. Also, I found in table 3 that, the larger-capacity model is less robust than the smaller-capacity model against white-box iterative attacks? This is strange.

---

> ### Author Response · Authors · 2018-11-27
> **Response to Reviewer 2**
>
> We address the concerns raised by the reviewer as follows.
>
> 1. The intuition behind using attention is to effectively assign weights to features depending on their robustness. Robust features are highly correlated with the class labels and invariant to input perturbations. Since the global features are directly used to produce class label prediction, we can use them as a query to assign attention weights. In this way, robust features that have higher correlations with class labels will be assigned larger weights which in turn contribute to the model's robustness. In order to validate this intuition, we conduct an additional experiment in Section 4.3 to examine the relationship between the robustness of a feature and its assigned attention weight. Figure 2 shows that more robust features are actually assigned larger weights by the attention module while the attention weights of non-robust features are small. In such a way, a model with attention has improved robustness.
>
> 2. We added the missing citations to our paper. For ALP, we agree that it is similar to our feature regularization approach. However, we argue that feature regularization is more intuitive than ALP. While the logits represent the prediction confidence of a model and pairing the logits prevents a model from being over-confident when making predictions, it's not entirely clear why this would lead to a more robust model. On the other hand, feature regularization motivates a model to learn very similar features for the clean and adversarial inputs. The learned features are invariant to input perturbations thus robust features. From another point of view, a model trained with feature regularization maps clean and adversarial examples from nearby points in the image space to nearby points in the high-dimensional manifold. We updated the related work section. The ImageNet results in ALP paper are still under development. Engstrom et al. (2018) invalidate their claims and the ALP paper was retracted from NIPS 2018 by the authors.
>
> Engstrom, Logan, Andrew Ilyas, and Anish Athalye. "Evaluating and understanding the robustness of adversarial logit pairing." arXiv preprint arXiv:1807.10272 (2018).
>
> 3. We have run the additional tests against the following adversaries: PGD 20+2, PGD 100+2, PGD 200+2, CW 30+2, CW 100+2 and updated our paper with the corresponding results in Section 4.2, Table 2. The results show that our proposed model is more robust than the baseline model against all adversaries in both white box and black box settings.
>
> 4. As suggested by the reviewer, we run the additional experiments with a wider ResNet to test our method. Due to time and resources constraints, we choose a 3-times wide ResNet with [16, 48, 96, 192] filters instead of the 10-times wide network in Madry et al. (2017). We believe experiments using a 3-times wide ResNet is able to demonstrate the effectiveness of our method with larger capacity networks and nevertheless, our method is independent of the model size. We present the results for this wide model on CIFAR-10 in Section 4.2, Table 3. From Table 3 we note that our method shows better performances against a wide range of adversaries than the baseline method.
>
> 5. We have run the experiments on CIFAR-100 and updated our paper with the corresponding results in Section 4.5, Table 5. The results exhibit a consistently better performance of our proposed model over the baseline method on a harder dataset like CIFAR-100.

---

> ### Author Response · Authors · 2018-12-04
> **Response to post-rebuttal comment**
>
> "Also, I found in table 3 that, the larger-capacity model is less robust than the smaller-capacity model against white-box iterative attacks? This is strange."
>
> - It's due to overfitting. Below is the record of the training and test accuracy relative to training epochs for wide networks against PGD-5. The first table is for the baseline model and the second table is for our model. It shows that while the training accuracy keeps increasing, the test accuracy first increases then decreases, which is a sign of model overfitting. Even that both models overfit, our method provides an improvement over the baseline method.
>
> Baseline model
> epoch                                      10              20             30              40              50             60             70            80            90            100
> training batch accuracy    52.34%     54.69%     59.38%     59.38%      65.62%     53.91%    56.25%    60.94%    56.25%     68.75%
> test accuracy                       43.86%     47.54%     49.67%     50.82%      50.63%     51.05%    50.66%    52.33%    51.84%     51.07%
> epoch                                    110            120           130           140            150           160            170         180          190           200
> training batch accuracy    78.91%     82.81%     78.12%     83.59%      79.69%     86.72%    88.28%    89.84%    92.19%     94.53%
> test accuracy                       54.29%     52.90%     51.81%     51.23%      50.68%     50.68%    49.98%    49.87%    49.03%     49.15%
>
> Our model
> epoch                                      10              20             30              40              50             60             70            80            90            100
> training batch accuracy    36.72%     49.22%     43.75%     51.56%      54.59%     56.01%    61.72%    54.21%    63.28%     70.31%
> test accuracy                      31.99%     38.22%     44.85%     47.82%      50.51%     51.27%    52.42%    52.16%    53.17%     53.26%
> epoch                                     110            120           130           140            150           160            170         180          190           200
> training batch accuracy   70.31%     61.38%     69.19%     66.28%      75.44%     85.59%    86.38%    91.41%    95.31%     93.75%
> test accuracy                      53.10%     52.68%     56.22%     54.38%      54.27%     54.64%    53.85%    53.71%    54.00%     53.23%

---

### Meta-Review · Area_Chair1 · 2018-12-14

**Confidence:** 4
**Recommendation:** Reject

**Metareview:**

The paper proposes an attention mechanism to focus on robust features in the
context of adversarial attacks. Reviewers asked for more intuition, more
results, and more experiments with different attack/defense models. Authors
have added experimental results and provided some intuition of their proposed
approach. Overall, reviewers still think the novelty is too thin and recommend
rejection. I concur with them.